# When Chromatin Decondensation Affects Nuclear γH2AX Foci Pattern and Kinetics and Biases the Assessment of DNA Double-Strand Breaks by Immunofluorescence

**DOI:** 10.3390/biom14060703

**Published:** 2024-06-14

**Authors:** Adeline Granzotto, Laura El Nachef, Juliette Restier-Verlet, Laurène Sonzogni, Joëlle Al-Choboq, Michel Bourguignon, Nicolas Foray

**Affiliations:** 1INSERM U1296 Unit “Radiation: Defense, Health, Environment”, Centre Léon-Bérard, 69008 Lyon, France; adeline.granzotto@inserm.fr (A.G.); laura.el-nachef@inserm.fr (L.E.N.); verlet@inserm.fr (J.R.-V.); laurene.sonzogni@inserm.fr (L.S.); joelle.al-choboq@inserm.fr (J.A.-C.); michel.bourguignon@inserm.fr (M.B.); 2Department of Biophysics and Nuclear Medicine, University Paris Saclay (UVSQ), 78035 Versailles, France

**Keywords:** DNA double-strand breaks, ATM, immunofluorescence, chromatin condensation, sodium butyrate

## Abstract

Immunofluorescence with antibodies against phosphorylated forms of H2AX (γH2AX) is revolutionizing our understanding of repair and signaling of DNA double-strand breaks (DSBs). Unfortunately, the pattern of γH2AX foci depends upon a number of parameters (nature of stress, number of foci, radiation dose, repair time, cell cycle phase, gene mutations, etc…) whose one of the common points is chromatin condensation/decondensation. Here, we endeavored to demonstrate how chromatin conformation affects γH2AX foci pattern and influences immunofluorescence signal. DSBs induced in non-transformed human fibroblasts were analyzed by γH2AX immunofluorescence with sodium butyrate treatment of chromatin applied after the irradiation that decondenses chromatin but does not induce DNA breaks. Our data showed that the pattern of γH2AX foci may drastically change with the experimental protocols in terms of size and brightness. Notably, some γH2AX minifoci resulting from the dispersion of the main signal due to chromatin decondensation may bias the quantification of the number of DSBs. We proposed a model called “Christmas light models” to tentatively explain this diversity of γH2AX foci pattern that may also be considered for any DNA damage marker that relocalizes as nuclear foci.

## 1. Introduction

By allowing the visualization of DNA damage sites in cell nuclei, immunofluorescence technique has revolutionized our view of the choreography of the actors of the DNA damage repair and signaling pathways in response to genotoxic stress, including ionizing radiation [1,2,3,4]. Immunofluorescence has also permitted the decrease of the threshold of drug concentration and radiation dose from which molecular events are measurable [5,6]. In particular, some proteins involved in the recognition, repair or signaling of DNA single-strand breaks (SSBs) or DNA double-strand breaks (DSBs) may form specific immunofluorescence spots (or foci) in the nucleus. This particular “relocalization” of these proteins as nuclear foci has been found to be due to their phosphorylation by the ataxia telangiectasia-mutated (ATM) or ataxia telangiectasia and Rad3 (ATR) kinases at the SSB or DSB sites. This is notably the case of checkpoint kinase 2 (CHEK2), breast cancer 1 (BRCA1), breast cancer 2 (BRCA2) proteins, mediator of DNA damage checkpoint protein 1 (MDC1), tumor suppressor p53-binding protein 1 (53BP1), meiotic recombination 11 (MRE11), DNA protein kinase (DNA-PK) and replication protein A (RPA) [1,2,3,4,7,8,9,10].

One of the most representative examples of proteins forming nuclear foci is the H2A histone X variant (H2AX) and its phosphorylated forms (γH2AX). H2AX is phosphorylated by some kinases like ATM, ATR and DNA-PK [8]. However, it is noteworthy that, unlike *ATM^-/-^* cells, *DNA-PK^-/-^* cells show early γH2AX foci [11]. The ATR-dependent γH2AX foci are generally observed after exposure to non-ionizing radiation like UV and some hours after exposure to ionizing radiation like X-rays, suggesting that ATM-dependent γH2AX foci represent an important subset in radiobiology [8,12]. The nuclear γH2AX foci have been shown to be early sensors of the DSBs managed by the non-homologous end-joining (NHEJ), the most predominant DSB repair pathway in humans and in quiescent mammalian cells [6,13]. In G2/M cells, the homologous recombination repair pathway is predominant even if some γH2AX foci may be observed, suggesting that NHEJ is activated even at this cell cycle phase. However, the nuclear γH2AX foci were also suggested to be linked to SSBs, even if this statement is not consensual, notably because the doses applied with the non-DSBs but SSB inducers were not biologically relevant [14]. A plethora of γH2AX data studies have contributed to advances in biodosimetry, basic knowledge of DNA repair and radiobiological characterization of genetic syndromes [5,11,15,16,17,18]. Anti-*γH2AX* immunofluorescence is therefore providing new insights in the evaluation of the quantity of genotoxic stress in very various experimental conditions, notably for doses as low as 1 mGy [6,19].

The H2AX histones show a rare specificity: they represent 2–25% of all the H2A histones structuring DNA, and they are unevenly distributed throughout chromatin [13,20,21,22,23]. Consequently, there are several potentially phosphorylable H2AX histones at the close vicinity of DSB sites: do the ATM, ATR and DNA-PK kinases phosphorylate all of them or only one H2AX molecule? The pattern of the resulting γH2AX foci is at the center of this question. Hence, while Rothkamm and Löbrich have shown that a one-to-one correlation exists between the number of γH2AX foci and that of DSBs [6], in practice, immunofluorescence shows nuclei with a large spectrum of different γH2AX foci patterns, which does not facilitate the analysis and the interpretation of data. It is noteworthy that such variety of γH2AX foci, already observed by a number of research groups, were not dependent on the microscopy technology applied (high-throughput microscopy systems, confocal microscope and/or simple immunofluorescence microscope) [16,24,25,26,27].

For more than twenty years, our laboratory has accumulated millions of γH2AX focus images in various experimental conditions of oxidative stress, i.e., ionizing radiation, UV and chemicals. The aim of this report is to better understand the changes to the nuclear H2AX foci patterns in different scenarios of exposure to stress and how stress may affect the assessment of DNA damage.

## 2. Materials and Methods

### 2.1. Cell Lines

This study was based on an analysis of several million γH2AX images accumulated by our laboratory over twenty years and obtained from several hundred human cell lines, including some lines from healthy tissues such as untransformed or transformed lung, intestine and skin fibroblasts, lymphoblasts, brain astrocytes and tumor cells, including sarcoma, glioma and carcinoma. Since we cannot provide the names or the biological features of the several hundred cell lines analyzed in our laboratory, we have deliberately chosen to select a subset of commercial cell lines tested in our laboratory that are very representative of the other cell lines. We have provided the biological features of the selected cell lines concerning the γH2AX images shown in the figures (Table 1). Cell lines were purchased from ScienCell Research Laboratories (Carlsbad, CA, USA), the European Collection of Authenticated Cell Cultures (ECACC, Salisbury, UK) or from American Type Culture Collection (ATCC, Manassas, VI, USA) (Table 1). With the notable exception of the cell cycle phase analysis, all the experiments were performed with cells in the plateau phase of growth (95–99% in G0/G1) to overcome any cell cycle effects. Cells were routinely cultured at 37 °C in 5% CO_2_ humid conditions as monolayers with Dulbecco’s modified Eagle’s minimum medium (DMEM) (Gibco-Invitrogen-France, Cergy-Pontoise, France), supplemented with 20% fetal calf serum, penicillin and streptomycin (Sigma-Aldrich, Saint-Quentin-Fallavier, France).

### 2.2. Chemicals

CuSO_4_ metal species (#451657), cis-platinum (cis-CDDP) (#232120), doxorubicin hydrochloride (#D1515), hydrogen peroxide (H_2_O_2_) (#H1009), and sodium butyrate (#5887) were purchased from Sigma-Aldrich, respectively. Drugs were added directly to the culture medium for the indicated times and at indicated concentrations [28].

### 2.3. Irradiation

Irradiation was performed with a 6 MeV X-ray medical irradiator (SL 15 Philips) (dose-rate: 6 Gy·min^−1^) at the Centre Léon-Bérard anti-cancer centre (Lyon, France) [29,30]. In all the experiments for which a γH2AX image is shown, a single dose of 2 Gy X-rays was applied. Dosimetry was certified by the radiophysics department of the Centre Léon-Bérard. Bystander conditions (medium change) consisted of the irradiation (10 Gy) of 1BR3 cells in the plateau phase of growth. Cells in their medium were centrifuged at 1500 rpm (300× *g*), and 10^6^ non-irradiated 1BR3 cells were incubated with the supernatant medium for 4 h at 37 °C. Donor and receptor cells were seeded in the same conditions [31]. For exposure to UV, a UVC (254 nm) chamber (#732-0859) from Fisher-Bioblock Scientific (Ilkirch, France) was used, and the fluence was routinely controlled by a UVC radiometer (Fisher-Bioblock; #UVPA97-0015-02) [32].

### 2.4. Immunofluorescence

Immunofluorescence protocol is described elsewhere [8,33]. Briefly, cells were fixed in 4% paraformaldehyde for 10 min at room temperature and were permeabilized in 0.5% Triton X-100 solution for 5 min at 4 °C. Such a permeabilization procedure has been found to avoid the production of any DNA breaks or artefactual foci in nuclei. Primary and secondary antibody incubations were performed for 1 h and 20 min at 37 °C, respectively. Anti-*γH2AX^ser^*^139^ antibody (#05636; Upstate Biotechnology-Euromedex, Mundolsheim, France) was used at 1:800. Anti-*53BP1* antibody (#05726; Upstate) was used at 1:100. Incubations with anti-mouse fluorescein (FITC) and rhodamine (TRITC) secondary antibodies were performed at 1:100 at 37 °C for 20 min. Slides were mounted on 4′,6′ Diamidino-2-Phenyl-indole (DAPI)-stained Vectashield (Abcys, Paris, France) for scoring micronuclei and mitoses and examined with a BX51 and ScanR (Olympus-France, Rungis, France) fluorescence microscope. DAPI staining also permitted us to indirectly evaluate the yield of G_1_ cells (nuclei with homogeneous DAPI staining), S cells (nuclei showing numerous γH2AX foci), G_2_ cells (nuclei with heterogeneous DAPI staining) and metaphase (visible chromosomes).

The focus scoring procedure applied here is certified by the CE mark and meets ISO-13485 quality management system norms. Our focus scoring procedure also has some features that are protected by the scope of the Soleau Envelop and are patented (FR3017625 A1, FR3045071 A1, EP3108252 A1) [33]. More than 50 nuclei per experiment were analyzed, and at least 3 independent replicates were performed for each condition [30]. The indicated concentrations of sodium butyrate (Sigma-Aldrich) were added directly to the culture medium immediately after irradiation or immediately after the lysis step. To consolidate our conclusion, it is noteworthy that when sodium butyrate was added after or between the antibody hybridization steps, no significant change in the pattern of γ-H2AX foci occurred when compared to adding butyrate after the lysis step. The quantification of integral fluorescence intensity and the size of foci were analyzed in 50 cells with the same length of exposure to fluorescence (500 ms in our conditions) and calculated with Cell^F^ v. 2015 software from Soft Imaging System GmbH (Münster, Germany) provided by the Olympus microscope. All the analyses were carried out, nucleus by nucleus, at ×100 magnification. Appendix A demonstrates that these analysis conditions were the best compromise between the average number of nuclei per field (at ×100 magnification, fewer than 10 nuclei per field according to their size) and the requirement of a magnification sufficient to score all the γH2AX minifoci.

### 2.5. Images and Statistical Analysis

The γH2AX kinetics data were fitted to the so-called Bodgi’s formula, which describes the kinetics of the appearance/disappearance of nuclear foci formed by some protein relocalizing after genotoxic stress. Statistical significance between data points was verified with one-way ANOVA. Statistical analysis was performed using Kaleidagraph v4 (Synergy Software, Reading, PA, USA).

## 3. Results

### 3.1. Current Observations with γH2AX Images: Seven Combinations of γH2AX Foci and Minifoci

In examining several million γH2AX images obtained over twenty years in our laboratory, two major types of foci were observed (Figure 1):-Some γH2AX foci showed a very intense immunofluorescence signal. The shape of these foci was not systematically round, and their average surface was found to be 5 ± 3 µm^2^ (from more than 10,000 different nuclei analyzed). When a large number (>100 foci) of these foci were present per cell, nucleus staining appeared uniform and intense.-Some γH2AX foci were about 10 to 20 times smaller than the previous ones and with a lower signal intensity, which made them difficult to detect and score. Their average surface was found to be 0.3 ± 0.1 µm^2^ (from more than 10,000 different nuclei analyzed). In this study, we called them “γH2AX minifoci”. When a very large number (>2000) of these minifoci were present, nucleus staining appeared sheepy and relatively intense (systematically less intense than a signal from the γH2AX foci described above).-If the two types of foci were absent from the immunofluorescence image, the background was dark (i.e., signal intensity was nil) (Figure 1).

From these above descriptions, seven major patterns were identified (Figure 1):-Nuclei with no background: neither γH2AX minifoci nor foci were observed (pattern I; e.g., non-irradiated fibroblasts from apparently healthy donors) or presence of γH2AX minifoci only (pattern II; e.g., non-irradiated fibroblasts from aging syndromes), or presence of γH2AX foci only (pattern III; e.g., non-irradiated fibroblasts from aging syndromes), or else presence of both γH2AX foci and minifoci (pattern IV; e.g., fibroblasts aging syndromes irradiated at low (mGy) doses).-Nuclei with numerous γH2AX minifoci (>2000 minifoci) forming a sheepy background: no γH2AX foci were observed (pattern V; e.g., non-irradiated fibroblasts from apparently healthy donors in S phase), or some γH2AX foci were observed (pattern VI; e.g., fibroblasts exposed to 30 µM CuSO_4_).-Nuclei with numerous γH2AX foci (>100 foci) forming an intense background (pattern VII; e.g., fibroblasts exposed to doses higher than 4 Gy).

The following representative examples were obtained:-With regard to the cell types: untransformed skin fibroblasts and astrocytes or even lymphoblasts generally elicited both γH2AX minifoci and foci. This statement was also relevant for tumor cells. However, the γH2AX minifoci were generally observed spontaneously in cells showing genomic instability and/or radiosensitivity and also shortly after exposure to stress (Figure 2A).-With regard to the cell cycle: The G1 cells showed both γH2AX minifoci and γH2AX foci, as described above. The S phase cells showed a large number of γH2AX minifoci. In the G2/M phase, the focus shape became more concentrated on chromatin and on chromosomes. In mitosis, the background was completely dark (Figure 2B).-With regard to the stress types: while 2 Gy-X-ray-irradiated fibroblasts showed a mixture of γH2AX foci and minifoci, fibroblasts exposed to UV, metal or bystander fibroblasts showed a majority of γH2AX minifoci and some rare γH2AX foci [28,31,32]. However, if UV dose or drug concentration increased drastically, the number of γH2AX foci increased, while the number of γH2AX minifoci decreased, at a rate depending on the nature of the stress induced. Interestingly, cisplatinum, which induces DNA adducts whose repair essentially involves SSBs, showed γH2AX minifoci [34], while bleomycin, a well-documented DSB inducer, elicited mostly γH2AX foci but very few γH2AX minifoci [35] (Figure 2C). Oxygen peroxide (H_2_O_2_) produces SSBs at low concentrations, and some DSBs appeared at high concentrations, leading to the γH2AX signal saturation (pattern VII) (Figure 2C). Altogether, these examples suggested a potential link between SSBs and DSB occurrence on one side and γH2AX minifocus and focus appearance on the other side.

### 3.2. Apparent Analogies between SSBs and DSBs vs. γH2AX Minifoci and Foci

In the case of γH2AX focus kinetics with human radioresistant fibroblasts exposed to X-rays: without irradiation, human radioresistant fibroblasts generally showed nuclei with a dark background, with fewer than 2 γH2AX foci on average and a number of γH2AX minifoci depending on the cell line (Figure 3A). Ten minutes after 2 Gy, nuclei showed a sheepy background with minifoci and about 80 γH2AX foci. Gradually, the number of γH2AX foci decreased progressively to reach 0–2 γH2AX foci at 24 h post-irradiation. At the same time, the number of γH2AX minifoci decreased faster than that of γH2AX foci, and no γH2AX minifoci were detected 1–4 h post-irradiation. In our experiments, the 2 Gy + 4 h scenario showed the most contrasted immunofluorescence image with a dark background and very intense γH2AX foci (Figure 3A). Interestingly, it is well documented that 2 Gy X-rays induce both SSBs and DSBs and that SSBs are repaired faster [32]: no SSBs appeared 4 h post-irradiation, while some residual DSBs may be fixed up to 24 h post-irradiation. Such observation suggested a strong temporal analogy between γH2AX foci and DSBs on one hand and γH2AX minifoci and SSBs on the other hand.

In the case of human radioresistant fibroblasts exposed to H_2_O_2_: treatment of cells with low concentrations of H_2_O_2_ resulted in the formation γH2AX minifoci (Figure 3B). As the H_2_O_2_ concentration increased, γH2AX foci appeared while the γH2AX minifoci disappeared. Interestingly, the literature shows that low concentrations of H_2_O_2_ lead to the formation of direct SSBs while increasing the concentrations of H_2_O_2_ leads to the formation of DSBs [36] (Figure 3B). Hence, the analogy established above between the DSBs and SSBs and the γH2AX foci and γH2AX minifoci, respectively, remains relevant with the treatment of H_2_O_2_. To investigate further this analogy, we summarized our observations in Table 2.

Table 2 suggests that:-regarding a dark background without any foci (pattern I), the induction of SSBs appeared with γH2AX minifoci under the condition that few DSBs are already present (patterns II and V) or that SSBs are so numerous that some DSBs are induced (patterns IV or VI). If these conditions are not met, the signal remains dark (pattern I). If repair is allowed, with the SSBs being repaired faster than DSBs, γH2AX minifoci disappeared faster than γH2AX foci.-regarding a dark background, the induction of DSBs was systematically associated with γH2AX foci (patterns III or VII). An increase of DSBs does not appear to be necessarily associated with SSBs: an increase in the number of γH2AX foci is not necessarily associated with an occurrence of γH2AX minifoci, except if the DSB repair involved SSBs such as with a recombination-like pathway [5,11,15,16,17,18]. In this case, the presence of minifoci can accompany each γH2AX foci (Table 2). Again, the analogy established above between the DSBs and SSBs and the γH2AX foci and minifoci, respectively, remains relevant with such treatments.

The fact that SSBs are likely to be reflected by γH2AX minifoci while DSBs are reflected by γH2AX foci does not necessarily mean that ATM kinase is activated by SSBs as it is by DSBs. By applying UVC to cells, Marti et al. observed γH2AX foci and suggested that ATM kinase was activated indifferently by SSBs or DSBs [14]. However, the dose of UVC applied in their study was so high that the probability of DSB induction by SSBs was not negligible [14]. Hence, how to reconcile the hypothesis that the number of γH2AX minifoci may reflect SSBs and γH2AX foci reflect DSBs if the ATM kinase is activated only by DSBs? (Table 2).

### 3.3. The Use of Sodium Butyrate to Decondense Chromatin

The scenarios presented in Table 2 suggested that the presence of some DSBs was required for the appearance of γH2AX minifoci: the induction of SSBs but not DSBs does not lead to a positive γH2AX signal, as if the phosphorylation of H2AX was triggered only in presence of DSBs but not of SSBs, which corresponds to a certain level of oxidative stress.

It must be stressed that the presence of SSBs also leads to chromatin decondensation. However, it is challenging to know whether γH2AX minifoci are directly due to the presence of SSBs or to the chromatin decondensation they induce. To address this question, preliminary experiments with some chromatin modifiers (but not SSB inducers) such as NaCl and sodium butyrate applied during the radiation treatment or after the cells fixation revealed drastic changes in γH2AX foci’s appearance [37,38,39]: in order to better estimate the impact of chromatin condensation upon the γH2AX focus pattern, we examined the effects of sodium butyrate on the most documented protocols that we applied in routine, namely the γH2AX focus kinetics of untransformed and radioresistant human fibroblast cell lines after a dose of 2 Gy X-rays that serve as a reference cellular model in our laboratory, notably the untransformed human 1BR3 fibroblasts [16].

The untransformed human 1BR3 fibroblasts were irradiated with 2 Gy X-rays, incubated at 37 °C for 10 min–24 h and pretreated with sodium butyrate and thereafter fixed as detailed in Materials and Methods. The great majority (97 ± 2%) of the unirradiated cells treated with sodium butyrate did not show any γH2AX foci or minifoci (pattern I), suggesting again that sodium butyrate does not induce any DNA break at the concentrations applied. Without sodium butyrate treatment, ten min after 2 Gy, 75 ± 6 γH2AX foci per cell were scored in agreement with other publications and with the average DSB induction rate of 37 ± 4 DSBs per Gy per human diploid quiescent cell. It is noteworthy that, under the same conditions, the SSB induction rate was shown to be about 1000 SSBs per Gy [15]. By increasing the sodium butyrate concentrations, the number of γH2AX foci progressively decreased in favor of γH2AX minifoci, whose number increased, while the number of SSBs increased with the radiation dose but not with the sodium butyrate concentration (Figure 4 and Figure 5). Such a trend with sodium butyrate treatment was observed at all the other post-irradiation times investigated (i.e., 1 h, 4 h, 24 h), while without sodium butyrate treatment, the γH2AX minifoci disappeared between 1 h and 4 h, as described above.

When γH2AX foci and corresponding minifoci were plotted together, an invariable feature of the foci was evident: the ratio between the two types of focus remains constant, independent of the repair time: the number of γH2AX minifoci was systematically found to be 13.67 ± 0.39 (average of the slopes of the curves shown in Figure 5F) times higher than the number of γH2AX foci (Figure 5F). These findings indicated that about 13 minifoci on average may compose a γH2AX focus and that chromatin decondensation (eventually due to sodium butyrate treatment or by the presence of indirect SSBs) may lead to the appearance of individual site of phosphorylation: the number of γH2AX minifoci increased and that of γH2AX foci decreased slowly (Figure 5F). To conclude, the formation of γH2AX minifoci requires few DSBs and a decondensed chromatin but not necessarily the formation of SSBs. Conversely, several SSBs may cause the appearance of γH2AX minifoci if such condition creates also few DSBs and chromatin decondensation. Considering that on average two molecules of H2AX are incorporated into every six nucleosomes, and the average nucleosome repeat length is about 200 bp, our findings are consistent with a DSB site that can cover 35–45 nucleosomes, corresponding to about 1–2 Mbp, in agreement with the literature [23].

## 4. Discussion

### 4.1. Evidence of the Variety of γH2AX Foci Patterns

The analysis of millions γH2AX focus images accumulated in the lab for more than twenty years revealed a wide variety of γH2AX foci patterns that, for some, do not facilitate analysis or focus scoring. Some other research groups have already pointed out such a variety [25,26,27]. Our γH2AX focus images were the results of a routine service consisting of the diagnosis of the individual radiosensitivity of cancer patients at the request of radiation oncologists to prevent severe reactions post-radiotherapy. This service has permitted our laboratory to assemble several hundred fibroblast cell lines with a broad spectrum of radiosensitivity, known as the COPERNIC collection [30]. A better knowledge of technical artefacts linked to the use of immunofluorescence was therefore necessary to interpret nucleus images and provide the best expertise possible [40].

It must be stressed here that our observations and our conclusions did not depend on the confocal nature of the microscope applied, the high-throughput rate of the system chosen, the color staining associated with primary and secondary antibodies or the cell fixation protocol [40]. Our scoring system based on three replicates and 50–100 nuclei analyzed per experiment appeared sufficient to achieve the best quality control (see Materials and Methods). Furthermore, focus scoring by eye may present a certain advantage on some high-throughput systems, since it drastically reduces the duration of the focus scoring time. As specified in Materials and Methods, we reached the certification level (CE and ISO levels) by comparing the focus scoring with several readers (eye scoring), several automatic systems and image analyzing software (ImageJ v1.8.0) [33] (see Materials and Methods).

By contrast, among the various technical features to consider, the choice of the lens magnification is crucial, since γH2AX minifoci are less visible than γH2AX foci. Hence, from our experience, we recommend the use of X100 magnification, which limits the use of high-throughput systems that are generally based on X60 magnification (the higher the magnification, the slower the data acquisition) (Appendix A). Moreover, in our experience, there was systematically a ratio equal to about 100/60 in the induction of γH2AX foci assessed per Gy per cell when comparing these two magnifications. Further investigations are therefore needed to better document the requirements of radiobiologists in focus scoring, and particularly γH2AX focus scoring, and establish arguments of choice far from the commercial propositions of the manufacturers. Lastly, it must be stressed that determining the difference between γH2AX foci and minifoci is not rigorously possible at ×60 magnification. Thus, for this reason, we have been obliged to present our images at ×100 magnification only (Appendix A).

In addition to the technical requirements for a rigorous analysis of the features of γH2AX and their scoring, some authors have recommended applying co-immunofluorescence with both 53BP1 and γH2AX markers to ensure that each focus represents one DSB [41]. We have provided evidence that (1) co-immunofluorescence may be artifactual [40] and that (2) 53BP1 focus kinetics may be very different from those obtained with the H2AX marker according to the cells tested and when these two biomarkers were applied separately [11]. Indeed, current fluorescent markers like the fluorescein isothiocyanate (FITC) tetramethyl rhodamine iso-thiocyanate (TRITC), elicit unavoidable overlaps of excitation and emission wavelength spectra (called bleed-through phenomena), independent of the nature of the microscopy approach (notably whether confocal or not) [40]. For example, we have shown previously that co-immunofluorescence of γH2AX and MRE11 biomarkers may lead to the conclusion that both proteins overlap from 10 min to 24 h post-irradiation, while the kinetics of their respective foci, obtained from separate one-marker immunofluorescence, revealed very different numbers of foci in the range time tested [40]. With regard to 53BP1, we have previously shown that, in human radioresistant fibroblasts, 2 Gy X-rays induce 15 53BP1 foci only 10 min post-irradiation, while 80 γH2AX foci were observed at the same time [11]. While the maximal number of 53BP1 foci was generally reached at 1 h, the maximal number of γH2AX foci was reached at 10 min post-irradiation, suggesting that 53BP1 and γH2AX cannot re-localize systematically when separate immunofluorescence is applied [11].

### 4.2. The Christmas Lights Model

The whole fluorescence signal emitted by nuclear foci is supposed to correspond to one protein molecule. However, H2A histone is a structural component of chromatin and H2AX variants represent 2–25% of H2A histone of the human genome [13,20,21,22,23]: several γH2AX molecules may be present in the close vicinity of the DSB sites, as long as the chromatin is condensed and these individual molecules are accessible to the ATM kinase. Here, a chromatin decondensing treatment applied after the irradiation and repair without producing DNA breaks resulted in the dispersion of these fluorescence H2AX molecules but kept the ratio between foci and minifoci constant (equal to 13.67 ± 0.39). For some cell lines (without SSBs and/or with highly condensed chromatin), the number of γH2AX minifoci is nil, which corresponds to a one-to-one correlation between DSBs and γH2AX foci, as already reported by Rothtkamm and Lobrich [6]. However, these authors have investigated only some radioresistant fibroblasts. In our experience, some fibroblast cell lines, derived from radiosensitive patients, do not obey this correlation and show a non-negligible number of γH2AX minifoci [16].

Our data suggest a model that we have called the Christmas lights model to tentatively explain the variation of γH2AX foci patterns and notably the existence of nuclear γH2AX minifoci. Let us consider the chromatin loops as a Christmas light chain in which each light corresponds to one H2AX variant histone protein. The given stress that induces DSBs on the loops leads ATM to phosphorylate all the H2AX histones in the close vicinity of the DSB site (on average, 13 H2AX molecules are involved). In the case of condensed chromatin, the planar projection of all the individual γH2AX positive signals concentrated around the DSB site results in the formation of one large and intense (standard) nuclear γH2AX focus. In these conditions, there is a one-to-one correlation between DSBs and γH2AX foci. In the case of a decondensed chromatin (that can be due to the presence of SSBs and some DSB formation, repair or other treatments like butyrate sodium that decondense chromatin without producing SSBs), the individual γH2AX positive signals move away from each other. Consequently, individual H2AX lights are more distinguishable and form an additional subpopulation of nuclear γH2AX minifoci, despite their being created by only one DSB. In such conditions, there is no one-to-one correlation between DSBs and γH2AX foci (Figure 6). Altogether, such a model illustrates the subtle balance between the consequences of DNA breaks formation on the chromatin state and the dispersion of H2AX molecules that collectively mark this formation.

## 5. Conclusions

The anti-*γH2AX* immunofluorescence technique is very promising. However, the enthusiasm generated by its impressive power and reliability must not hide the existence of some technical artefacts that may bias quantitative and qualitative data interpretation. Here, we stressed the fact that intrinsic or stress-induced chromatin decondensation may disperse the fluorescence γH2AX signal, increase the apparent number of foci, and change their shape and brilliance. Such artefacts particularly concern flow cytometry techniques involving the γH2AX signals because they are based on the assessment of the whole fluorescence signal, and obviously immunofluorescence applied to cells [18]. Furthermore, the settings of analysis software should take into account the γH2AX minifoci, whose number may provide helpful information about the chromatin state before nuclei fixation. To conclude, in order to quantify precisely the actual number of DSBs induced by a given experimental protocol (and not the apparent number of DSBs), it appears therefore crucial to consider any γH2AX data together with the chromatin state induced by the experimental protocol used and to accumulate quantitative information about the different foci patterns.

## 6. Patents

WO2017029450–Individual method predictive of the DNA-breaking genotoxic effects of chemical or biochemical agents.

## Figures and Tables

**Figure 1 biomolecules-14-00703-f001:**
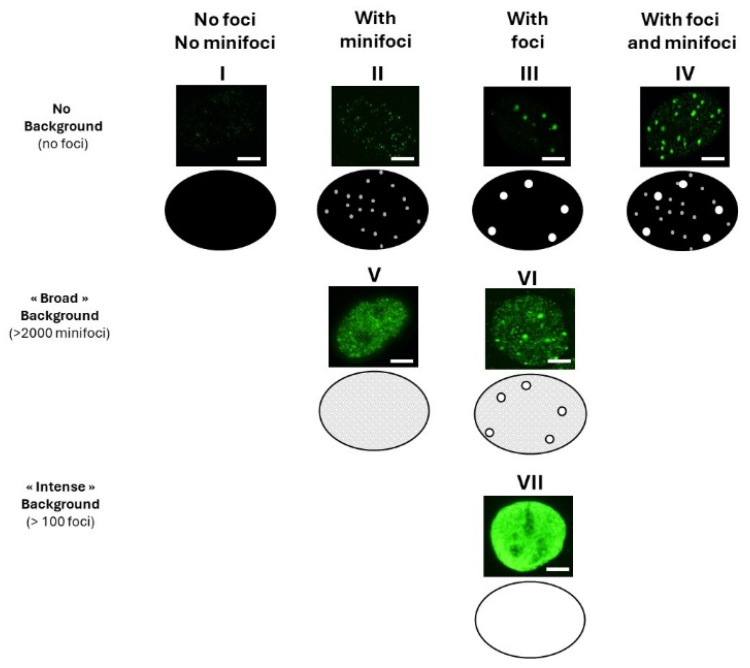
The seven representative patterns of γH2AX focus images obtained from the database of our laboratory from the 7 representative cell lines described in Materials and Methods. White bar represents 5 µm.

**Figure 2 biomolecules-14-00703-f002:**
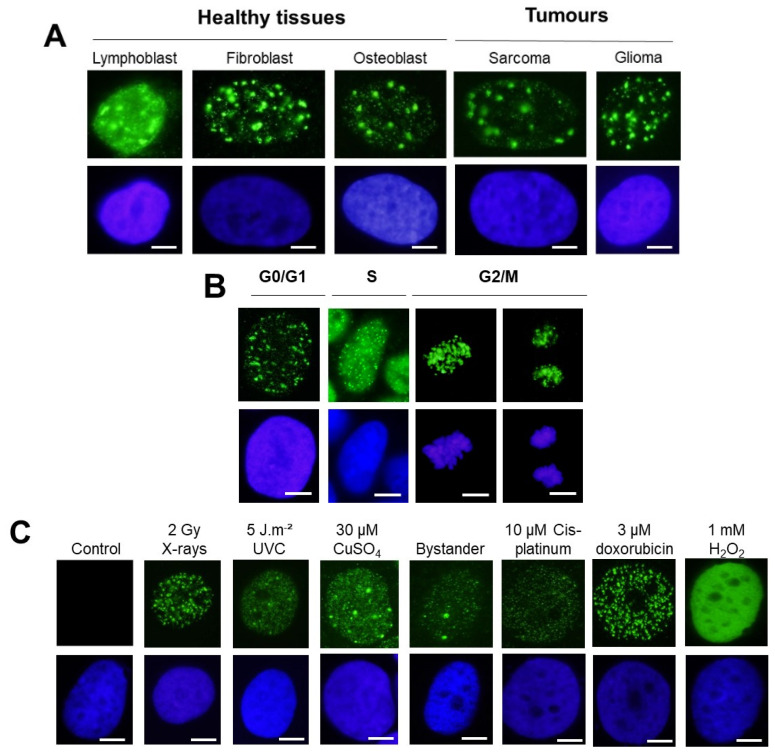
**(A**). Representative examples of patterns of γH2AX focus images obtained from different quiescent human cell types (as indicated) irradiated at 2 Gy X-rays followed by 1 h for repair. (**B**). Representative examples of patterns of γH2AX focus images obtained from untransformed human fibroblasts at the indicated cell cycle phases after 2 Gy X-rays followed by 1 h for repair. (**C**). Representative examples of patterns of γH2AX focus images obtained from untransformed quiescent human fibroblasts exposed to different stresses as indicated: 2 Gy X-rays followed by 1 h for repair, 5 J·m^−2^ UVC followed by 1 h for repair, 30 µM CuSO_4_ for 24 h, and exposed for 4 h to cell culture medium in which 1 million cells were exposed to 10 Gy followed by 1 h for repair, 10 µM cis-platinum for 24 h, 3 µM doxorubicin for 30 min and 1 mM H_2_O_2_ for 15 min. For all the panels, nuclei were counterstained with DAPI. White bar represents 5 µm.

**Figure 3 biomolecules-14-00703-f003:**
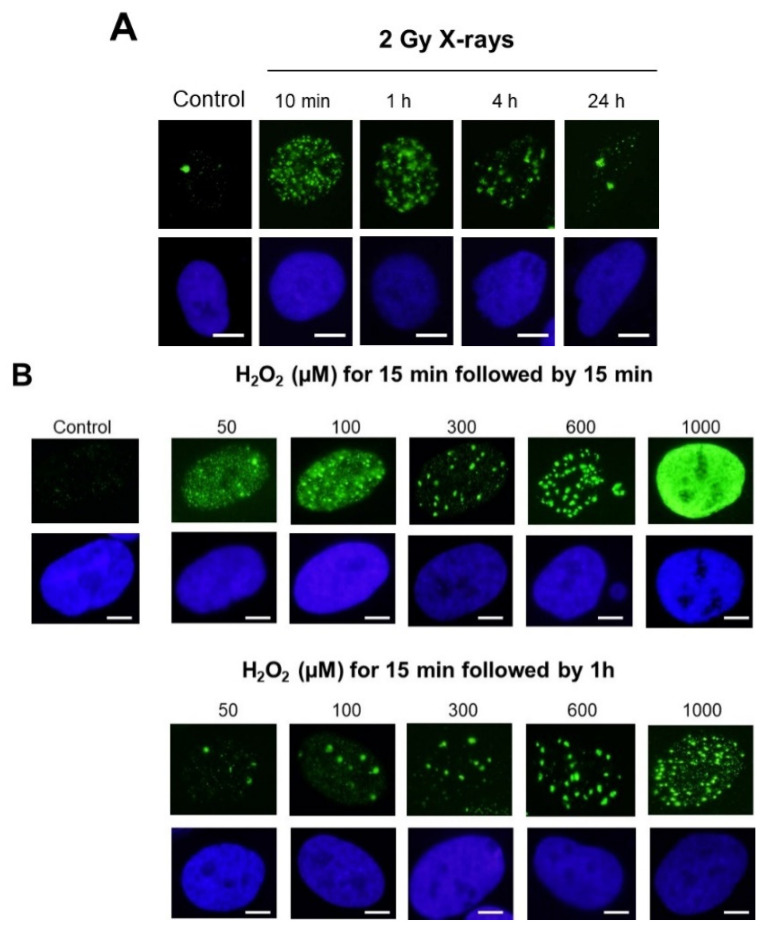
(**A**). Representative examples of γH2AX focus images obtained from the radioresistant human 1BR3 fibroblast cell lines exposed to 2 Gy X-rays followed by the indicated post-irradiation times for repair. (**B**). Representative examples of γH2AX focus images obtained from the radioresistant human 1BR3 fibroblast cell lines exposed to the indicated concentrations of hydrogen peroxide (H_2_O_2_) followed by 15 min or 1 h for repair. For all the panels, nuclei were counterstained with DAPI. White bar represents 5 µm.

**Figure 4 biomolecules-14-00703-f004:**
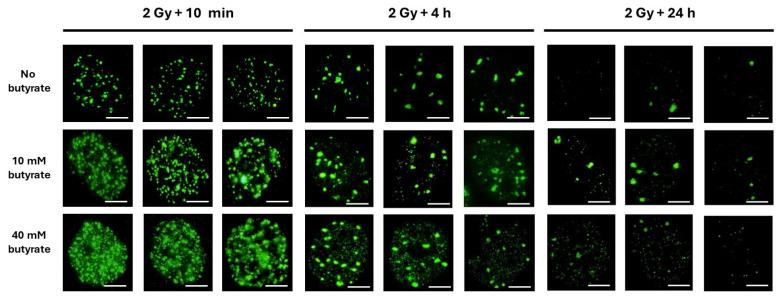
Three representative examples of γH2AX focus images obtained from the radioresistant human 1BR3 fibroblast cell lines exposed to 2 Gy X-rays, followed by the indicated post-irradiation times for repair and treated with the indicated concentration of sodium butyrate applied immediately after the period of r data are expressed in percentages, the γH2AX minifocus kinetics were found to be similar to those in pre-treated cells (*p* > 0.6). The γH2AX minifocus kinetics of non-pre-treated cells revealed a faster rate of focus disappearance than those of pre-treated cells (*p* < 0.04) but a slower rate than the SSB repair rate (*p* > 0.3). These findings suggested that γH2AX minifoci reflect the effects of the chromatin decondensation induced by sodium butyrate but cannot reflect each of the radiation-induced SSBs (Figure 5C,D). When expressed as a function of sodium butyrate concentration, the number of γH2AX minifoci increased, while the number of radiation-induced DNA breaks (whether DSBs or SSBs) are not seen to vary with sodium butyrate treatment. Furthermore, the number of SSBs after 2 Gy X-rays after 10 min was expected to be about 2000 per nucleus, while the number of γH2AX minifoci did not exceed 400. At 1 h post-irradiation, the radiation-induced SSBs had disappeared, while the number of γH2AX minifoci reached about 100 after 40 mM treatment (Figure 5E).

**Figure 5 biomolecules-14-00703-f005:**
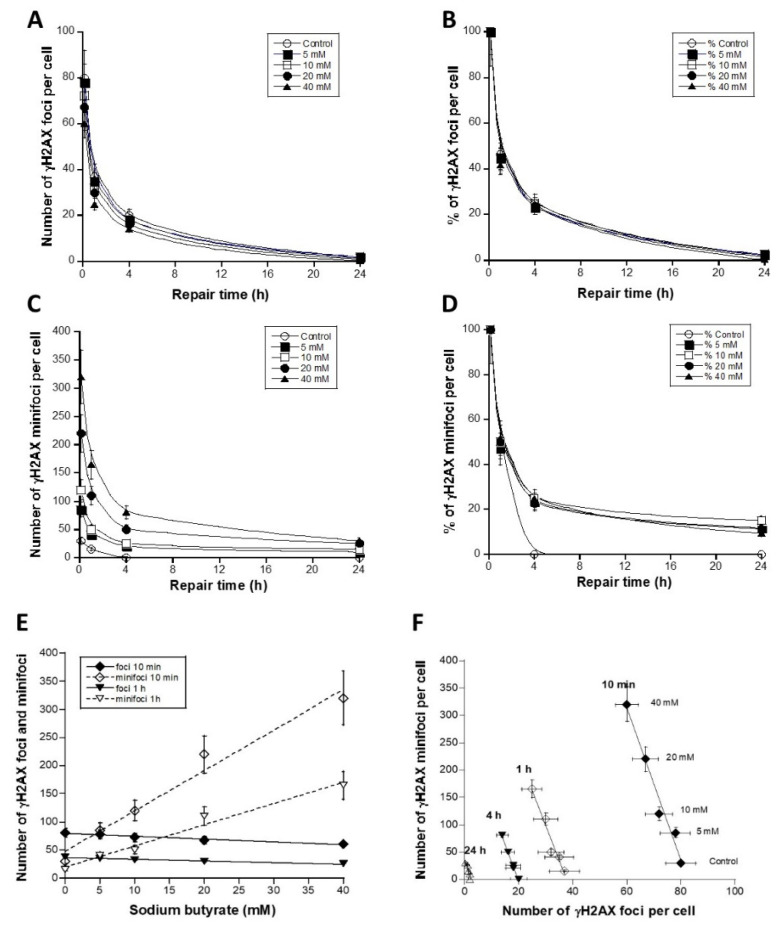
(**A**). Number of γH2AX foci per cell obtained from the human radioresistant 1BR3 fibroblast cell lines exposed to 2 Gy X-rays followed by the indicated post-irradiation times for repair and treated with the indicated concentration of sodium butyrate. Each plot represents the mean ± standard error of the mean (SEM) of 3 independent replicates. (**B**). The same data shown in panel A but expressed as percentages (data were divided by the corresponding number of γH2AX foci per cell 10 min post-irradiation). (**C**). Number of γH2AX minifoci per cell obtained in the same experimental conditions as described in panels A and B. Each plot represents the mean ± standard error of the mean (SEM) of 3 independent replicates. (**D**). The same data shown in panel (**C**) but expressed as percentages (data were divided by the corresponding number of γH2AX minifoci per cell 10 min post-irradiation). (**E**). Numbers of γH2AX foci and minifoci per cell shown in panels (**A**,**C**) expressed as a function of sodium butyrate. (**F**). The numbers of γH2AX foci (shown in panel (**A**)) were plotted against the corresponding numbers of γH2AX minifoci (shown in panel (**C**)). Each plot represents the mean ± standard error of the mean (SEM) of 3 independent replicates. For each repair time from 10 min to 24 h, data were fitted to the linear functions, respectively: y = 1150 − 13.94x; r = 0.98; y = 480 − 12.72x; r = 0.97; 266 − 13.46x; r = 0.99; y = 35.8 − 14.57x; r = 0.92.

**Figure 6 biomolecules-14-00703-f006:**
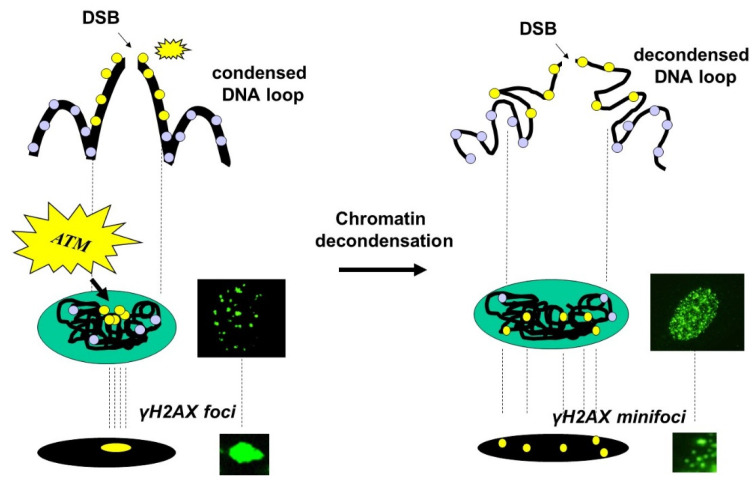
The Christmas lights model. Schematic illustration of the effect of chromatin decondensation on the formation of γH2AX foci.

**Table 1 biomolecules-14-00703-t001:** Major features of the cell lines whose nuclei are shown in this study.

Cell Lines	Nature of the Tissue	Radiosensitivity	Collection
Healthy tissue cells lines			
MRC5	Untransformed skin fibroblast	Apparently healthy patient	ECACC
1BR3	Untransformed skin fibroblast	Apparently healthy patient	ECACC
144BR	Untransformed skin fibroblast	Apparently healthy patient (same as LB173)	ECACC
LB173	Epstein–Barr virus (EBV)-transformed lymphoblastoid cells	Apparently healthy patient (same as 144BR)	ECACC
HA	Fetal cortex astrocytes		Sciencell (#1800)
Tumor cell lines			
U2OS	Osteosarcoma		ATCC (#HTB-96)
U118	Glioblastoma		ATCC (#HTB-15)

**Table 2 biomolecules-14-00703-t002:** Link between SSBs and DSBs and γH2AX minifoci and foci.

	SSB-Induced	DSB-Induced	γH2AX Minifoci	γH2AX Foci
**Cell type**				
Lymphoblasts	x	x	x	x
Fibroblasts	x	x	x	x
Osteoblasts	x	x	x	x
Sarcomas	x	x	x	x
Gliomas	x	x	x	x
**Cell cycle phase**				
G0/G1	x	x	I, II, V	I, III, IV, VI, VII
S	x		II, V	
G2/M	x	x		x
**Stress type**				
Control	0	0	I	I
X-rays				
UVC	x	x ^a^	II, IV, V, VI	III, IV, VI, VII
CuSO_4_	x	x ^a^	II, IV, V, VI	III, IV, VI, VII
Bystander	x	x ^a^	II, IV, V, VI	III, IV, VI
Cisplatinum	x		II, V, VI, VII	
Doxorubicin		x		III, VII
H_2_O_2_	x	x ^a^	II, IV, V, VI	III, IV, VI, VII

^a^ at high concentrations.

## Data Availability

All the data can be provided upon reasonable request.

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
