# Peer review of "When Chromatin Decondensation Affects Nuclear γH2AX Foci Pattern and Kinetics and Biases the Assessment of DNA Double-Strand Breaks by Immunofluorescence"

_biomolecules, 2024, doi:10.3390/biom14060703_

Round 1
Reviewer 1 Report
Comments and Suggestions for Authors
In the manuscript, "When chromatin deconsdensation affects nuclear gH2AX foci pattern and kinetics and biases assessment of DNA double strand breaks by immunofluorescence", the authors examine how chromatin status affects the status of H2AX phosphorylation after genotoxic stress.
This is rather interesting manuscript as it examines possible complications in an assay that is used routinely in the DNA repair field.
The data seems quite sound.
The manuscript is well written and easy to understand.
All figures are required.
Minor comments, Figures 2, 3 and 4 all have images with a single cell. Images with multiple cells should be used for each panel. The reviewer understands that the authors used 100X magnification for studies, but multiple cells should still be possible.
The authors should examine additional DNA repair proteins, such as 53BP1.
Major comment, there appears to be a lack of rigor in the current study. All the data presented appears to be produced in a single cell line. These question comes up is does these observations hold up in multiple cell lines. Seeing the data in additional cell lines would strengthen the authors conclusions.
Author Response
Reply to reviewer 1 :
We thank the reviewer for his/her comments
In the manuscript, "When chromatin deconsdensation affects nuclear gH2AX foci pattern and kinetics and biases assessment of DNA double strand breaks by immunofluorescence", the authors examine how chromatin status affects the status of H2AX phosphorylation after genotoxic stress.
This is rather interesting manuscript as it examines possible complications in an assay that is used routinely in the DNA repair field.
The data seems quite sound.
The manuscript is well written and easy to understand.
All figures are required.
Minor comments, Figures 2, 3 and 4 all have images with a single cell. Images with multiple cells should be used for each panel. The reviewer understands that the authors used 100X magnification for studies, but multiple cells should still be possible.
OK By considering the image size reduction to the journal format and the X100 magnification that is absolutely required to better visualize and quantify minifoci, the only format possible for a rigorous analysis was to screen nucleus by nucleus at X100. In order to convince the reviewer,we have added a figure in supplementary data with different magnifications to show the multiple cells fields and the consequence of a reduction of size. See modified text and figure in Materials and methods and in discussions.
The authors should examine additional DNA repair proteins, such as 53BP1.
OK We are aware that some reports encourage the application of co-immunofluorescence with 53BP1 and H2AX to visualize DSB. However, we have provided evidence that 1) co-immunofluorescence may be artifactual and 2) 53BP1 foci kinetics may be very different from those btained with the H2AX marker in human cells when these two biomarkers are applied separately. See modified text in Discussion and additional figure in supplementary data.
Major comment, there appears to be a lack of rigor in the current study. All the data presented appears to be produced in a single cell line. These question comes up is does these observations hold up in multiple cell lines. Seeing the data in additional cell lines would strengthen the authors conclusions.
See the reply to the 1st comment above. We stressed the fact that details in Materials and Methods demonstrate that we cannot provide million of images and even several cells per fields to nbetter quantify minifoci. To the opposite of the comment, it is a sign of rigor to have examined H2AX features nucleus by nucleus at 100X magnification : at X60, our conclusions are impossible to reach. See modified text and figure in Materials and methods.
Reviewer 2 Report
Comments and Suggestions for Authors
In this manuscript, Granzotto et al. sought to explore and characterize two types of γH2AX foci - small, low intensity minifoci and large foci showing intense IF signal. As the analysis of γH2AX foci levels is routinely used as a readout for DNA damage induction and repair in the cells, the topic is in principle interesting, important and suitable for publication. However, this paper has several serious drawbacks that, in my opinion, preclude its publication in the present form.
Major points:
A) In my view this paper somewhat lacks originality. Presence of different γH2AX foci (small and large) has been noticed more than 15 years ago. These foci were also characterized (at least to some degree). Further, different patterns of γH2AX foci appearance, that depend on cell cycle stage, were also reported earlier (see for example McManus and Hendzel, 2005, Mol Biol of the Cell; Ismail and Hendze, 2008, Envir and Mol Mutagenesis, Rybak et al. 2016, Oncotarget). Finally, there are convincing, previously published results, linking chromatin condensation and decondesation with γH2AX foci formation/disassembly (e.g. Mazumdar et al.; 2006; Curr Biol; Burgess et al.; 2014; Cell Reports).
B) Throughout the manuscript Authors suggest that γH2AX minifoci mark chromatin fragments with SSB’s while large γH2AX foci represent chromatin locations suffering from DSBs. However, they do not show any direct evidence beside simple correlation between minifoci appearance/disassembly and treatment with selected DNA damage agents. If minifoci indeed represent SSB and foci DSB it should be directly shown e.g. by comet assay or new generation sequencing-related techniques. Samples can be taken at indicated time points when one type of DNA damage is still present (DSB) while the other (SSB) is being largely repaired.
C) Moreover, it was earlier shown that small γH2AX foci found in untreated cells do not localize with checkpoint or repair proteins while large foci do (McManus and Hendzel, 2005, Mol Biol of the Cell). This raises a possibility that large fraction of γH2AX minifoci do not mark DNA break sites. Therefore, the statement that small γH2AX minifoci are SSB can be highly inaccurate. To clarify this issue, Authors should immunofluorescently label cells not only with anti-γH2AX but also using other Ab’s raised against DNA repair proteins (e.g. PARP, RAD51).
Minor points:
A) There are evidence that subtelomeric regions can be naturally enriched in γH2AX. Therefore, some of the minifoci might in fact mark these genomic regions. Also, few paper show that many of the γH2A.X foci found in tumor cells decorate disrupted telomeres.
B) Did Authors consider a possibility that at least some of the γH2AX foci are artefacts resulting from cell fixation and permeabilization procedure? Or off-target binding of γH2AX antibody? (see for example Schnell et al.; 2014; Nature Methods).
Comments on the Quality of English LanguageSome proof-reading required.
Author Response
Reply to reviewer 2 :
We thank the reviewer for his/her comments
In this manuscript, Granzotto et al. sought to explore and characterize two types of γH2AX foci - small, low intensity minifoci and large foci showing intense IF signal. As the analysis of γH2AX foci levels is routinely used as a readout for DNA damage induction and repair in the cells, the topic is in principle interesting, important and suitable for publication. However, this paper has several serious drawbacks that, in my opinion, preclude its publication in the present form.
Major points:
In my view this paper somewhat lacks originality. Presence of different γH2AX foci (small and large) has been noticed more than 15 years ago. These foci were also characterized (at least to some degree). Further, different patterns of γH2AX foci appearance, that depend on cell cycle stage, were also reported earlier (see for example McManus and Hendzel, 2005, Mol Biol of the Cell; Ismail and Hendze, 2008, Envir and Mol Mutagenesis, Rybak et al. 2016, Oncotarget). Finally, there are convincing, previously published results, linking chromatin condensation and decondesation with γH2AX foci formation/disassembly (e.g. Mazumdar et al.; 2006; Curr Biol; Burgess et al.; 2014; Cell Reports).
We fully agree with the reviewer and these references have been now included in the modified text. However, most of these references consider chromatin decondensation via the induction of DNA damage and our conclusions, compatible with these reports, bring a new and more quantitative vision of the role of gH2AX foci, dependent on chromatin decondensation but not on the presence of SSB:
- McManus and Hendzel (2005) concluded that small foci do not recruit any DSB, which is consistent with our conclusion but they more likely focused on cell cycle and ATR-dependent event to explain the existence of small γH2AX Here, with butyrate and H2O2 data, we have brought the hypothesis of occurrence of minifoci by chromatin decondensation without creating SSB.
- Ismail and Hendzel (2008) and Rybak et al. 2016, stated that H2AX are not necessarily a marker of DSB by arguing that with UV or H202, considered by the authors as non-DSB inducer, there are H2AX foci. In this paper, we provided evidence that the huge (and not biologically relevant) doses of UV applied produce DSB indirectly and that increased concentration of H2O2 are known to produce SSB and thereafter DSB. Finally they argued that the presence of DSB should be verified by a co-localization of H2AX and 53BP1 markers : we have a significant experience of technical artifacts linked to the co-immunofluorescence and the fact that 53BP1 foci kinetics may be very different from the kinetics of H2AX foci when these biomarkers are applied separately. See modified text and in Discussion.
- Mazumdar et al (2006) and Burgess et al 2014 investigated the consequences of the chromatin decondensation induced by DNA damage (DSB , eventually caused by siRNA). Here, it must be stressed that the sodium butyrate induce a chromatin decondensation without creating DNA breaks.
See also below
Throughout the manuscript Authors suggest that γH2AX minifoci mark chromatin fragments with SSB’s while large γH2AX foci represent chromatin locations suffering from DSBs. However, they do not show any direct evidence beside simple correlation between minifoci appearance/disassembly and treatment with selected DNA damage agents. If minifoci indeed represent SSB and foci DSB it should be directly shown e.g. by comet assay or new generation sequencing-related techniques. Samples can be taken at indicated time points when one type of DNA damage is still present (DSB) while the other (SSB) is being largely repaired.
There is a clear misunderstanding through this comment. In fact, we did not state that each minifoci represents SSB. We showed that the presence of minifoci was observed at each time that SSB and some DSB are present. However, the butyrate sodium treatment that decondenses chromatin but that does not create SSB or DSB reveals the presence of minifoci. Hence, it is the chromatin decondensation whether due or not to SBB that induced minifoci. As a consequence, the ratio between large foci and minifoci is constant while if SSB was the only condition of occurrence of minifoci, this ratio would be variable. See modified text to better stress this fact all along the paper.
Moreover, it was earlier shown that small γH2AX foci found in untreated cells do not localize with checkpoint or repair proteins while large foci do (McManus and Hendzel, 2005, Mol Biol of the Cell). This raises a possibility that large fraction of γH2AX minifoci do not mark DNA break sites.
Yes we agree but again our conclusions are not that minifoci is created by SSB.. notwithstanding the fact that co-immunofluorescence may be artefactual. See also modified text about the quantitative features of minifoci and those related to other DNA repair and signaling proteins. Again, our conclusion is that γH2AX minifoci are dependent on the chromatin condensation but not on the SSB incidence, as our data with butyrate sodium suggested it.
Therefore, the statement that small γH2AX minifoci are SSB can be highly inaccurate.
Again, we did not state that at all. See above.
To clarify this issue, Authors should immunofluorescently label cells not only with anti-γH2AX but also using other Ab’s raised against DNA repair proteins (e.g. PARP, RAD51).
The scope of this paper has been clearly identified as the technical artefacts related to γH2AX immunofluorescence. H2AX is a constituent of DNA scaffold while PARP and RAD51 are DNA single-strand breaks repair proteins. The artefacts observed with H2AX are not necessarily similar to those observed with DNA repair proteins. Furthermore, as written above, co-immunofluorescence with H2AX and PARP or H2AX and RAD51 may be also artefactual and very likely cell cycle dependent: unlike G2/M cells, RAD51 foci do not exist in G0/G1 cells. Similarly, there is no quantitative agreement with PARP foci observed in S phase and H2AX foci in the same condition.
Minor points:
- There are evidence that subtelomeric regions can be naturally enriched in γH2AX. Therefore, some of the minifoci might in fact mark these genomic regions. Also, few paper show that many of the γH2A.X foci found in tumor cells decorate disrupted telomeres.
Again, if cells, like cells from aging syndromes, show spontaneous SSB and decondensed chromatin, the minifoci may appear. For example, in our hands, fibroblasts from the progeria, Werner’s syndrome and Cockayne’s syndromes show both spontaneous SSB, minifoci and decondensed chromatin. As far as subtelomeric regions hold SSB that lead to chromatin decondensation, some minifoci could be observed. But, again here, we have associated the minifoci to chromatin decondensation independently of the presence of SSB. This question appears therefore out of the scope of the paper. However, there is another question raised by the reviewer about the enrichment : the telomeric sequences (TTACGG)n do not correspond to an enrichment in SQ or TQ domains (like H2AX). Therefore, the enrichment is not an “intrinsic” enrichment of the DNA sequence in H2AX but rather an enrichment in SSB because of the accessibility of the telomeres. Hence, the statement that H2AX represent 2-25% of H2A histone of the human genome may be also relevant for telomeres and therefore such specific region should also obey the ratio of about 13 between γH2AX foci and minifoci.
Did Authors consider a possibility that at least some of the γH2AX foci are artefacts resulting from cell fixation and permeabilization procedure? Or off-target binding of γH2AX antibody? (see for example Schnell et al.; 2014; Nature Methods).
The paraformaldehyde procedure is applied to the lab since 20 years since it permits the best compromise between a good fixation allowing antibodies to bind to antigen and avoid any artefactual hole in DNA or additional foci like for example acetone. Hence, the production of artefactual foci is theoretically possible but their number will be completely negligible with regard the number of radiation induced ones. Anyway, we have experience and routine procedures to verify such artefact. See modified text in Materials and Methods. (The reference Schnell et al is of 2012 and not 2014).
Reviewer 3 Report
Comments and Suggestions for Authors
The authors investigated phosphorylated histone H2AX mark in the context of chromatin with various levels of condensation and in many different cell types. The study is relevant as gH2AX is widely used as a marker of DNA damage and DNA repair, although as an indirect marker. Limitations of this marker, including technical ones, are important to highlight for the development of the research field.
Specific points:
1. Kindly spell abbreviations when used for the first time. Although clear for specialists, it is a tradition to introduce specific terms when first used for a broader audience. For example, ATM, ATR, CHK2, BRCA1/2, MDC1, 53BP1, MRE11, RPA, etc.
2. H2AX is known to be phosphorylated by several kinases, not just an ATM (line 49). Kindly introduce that H2AX can be phosphorylated at least by ATM, ATR, and DNA-PK.
3. Although DSBs are managed by NHEJ, they are also managed by HR (and A-EJ). Kindly mention it in the introduction. Otherwise, the readers will be left with the impression that DSB and gH2AX are related only to NHEJ.
4. It is confusing to read that the study is based on several hundreds of cell lines, and then to see a Table 1 with 7 lines only. Kindly fix this discrepancy in the Methods (and paper overall). If the study is based on these 7 lines, do not mention hundreds of lines available. If indeed hundreds of lines were used, kindly list all of them (relevant for the study of course), including origin, as not all of them might be commercially available.
5. Line 110 mentions that the cells were centrifuged at 1500 rpm. This value is not informative as "rpm" will result in different outcomes depending on the rotors and centrifuges. Kindly provide information about the centrifuge and rotors used, and, in addition to 1500 "rpm", kindly indicate expected "g".
6. Line 113. UVC chamber is mentioned. If possible, kindly specify expected wavelength produced by the lamp in this chamber. Was it monitored with a detector? Which detector, if any? What is the range of wavelengths for UV-C (indicate for readers)? Was it 100nm? or 280? or a mixture?
7. Section 3.1. and Figure 1. Kindly indicate examples of cells and treatments for each presented type. Discuss why in some cases there is nearly no signal, and in some cases there is too much signal. Is it because of no DNA damage and extensive DNA damage? This can be left for Discussion section, but briefly, basic information can be included in section 3.1.
Overall, the paper is a nice collection of useful summaries related to gH2AX staining in human cells at various conditions, based on the decades of the researchers' expertise.
Author Response
Reply to reviewer 3 :
We thank the reviewer for his/her comments
The authors investigated phosphorylated histone H2AX mark in the context of chromatin with various levels of condensation and in many different cell types. The study is relevant as gH2AX is widely used as a marker of DNA damage and DNA repair, although as an indirect marker. Limitations of this marker, including technical ones, are important to highlight for the development of the research field.
Specific points:
Kindly spell abbreviations when used for the first time. Although clear for specialists, it is a tradition to introduce specific terms when first used for a broader audience. For example, ATM, ATR, CHK2, BRCA1/2, MDC1, 53BP1, MRE11, RPA, etc.
OK see modified text in Introduction lines 46-57
- H2AX is known to be phosphorylated by several kinases, not just an ATM (line 49). Kindly introduce that H2AX can be phosphorylated at least by ATM, ATR, and DNA-PK.
OK see modified text in Introduction lines 52-73
- Although DSBs are managed by NHEJ, they are also managed by HR (and A-EJ). Kindly mention it in the introduction. Otherwise, the readers will be left with the impression that DSB and gH2AX are related only to NHEJ.
OK see modified text in Introduction lines 53-62. The mention of the different NHEJ variants are included in the new reference cited.
- It is confusing to read that the study is based on several hundreds of cell lines, and then to see a Table 1 with 7 lines only. Kindly fix this discrepancy in the Methods (and paper overall). If the study is based on these 7 lines, do not mention hundreds of lines available. If indeed hundreds of lines were used, kindly list all of them (relevant for the study of course), including origin, as not all of them might be commercially available. This is very important for us to mention that the features of gH2AX foci were observed undifferently in all the cell lines investigated in our lab. See the modified text proposed in Materials and Methods lines 97-100
- Line 110 mentions that the cells were centrifuged at 1500 rpm. This value is not informative as "rpm" will result in different outcomes depending on the rotors and centrifuges. Kindly provide information about the centrifuge and rotors used, and, in addition to 1500 "rpm", kindly indicate expected "g".
OK see modified text in materials and Methods line 126
- Line 113. UVC chamber is mentioned. If possible, kindly specify expected wavelength produced by the lamp in this chamber. Was it monitored with a detector? Which detector, if any? What is the range of wavelengths for UV-C (indicate for readers)? Was it 100nm? or 280? or a mixture?
OK see modified text lines 128-130
- Section 3.1. and Figure 1. Kindly indicate examples of cells and treatments for each presented type. Discuss why in some cases there is nearly no signal, and in some cases there is too much signal. Is it because of no DNA damage and extensive DNA damage? This can be left for Discussion section, but briefly, basic information can be included in section 3.1.See modified text in section 3.1
Overall, the paper is a nice collection of useful summaries related to gH2AX staining in human cells at various conditions, based on the decades of the researchers' expertise.
Round 2
Reviewer 1 Report
Comments and Suggestions for Authors
This is a revised version of manuscript submitted. The authors have addressed previous concerns. This paper is acceptable.
Author Response
We thank the reviewer for his/her comments.
Reviewer 2 Report
Comments and Suggestions for Authors
I would like to thank the authors for addressing my comments. Not all of my concerns have been fully resolved, but the manuscript has been sufficiently improved that I can recommend its publication.
Comments on the Quality of English LanguageThe newly written sections need to be revised - they contain quite a few typos.
Author Response
I would like to thank the authors for addressing my comments. Not all of my concerns have been fully resolved, but the manuscript has been sufficiently improved that I can recommend its publication.
The newly written sections need to be revised - they contain quite a few typos.
We thank the reviewer for his/her comments.
See modified text and typos corrected